# Learning Modular Simulations for Homogeneous Systems

**Jayesh K. Gupta**\*, **Sai Vemprala**\*, **Ashish Kapoor**
Microsoft Autonomous Systems and Robotics Research
`<jayesh.gupta,savempra,akapoor>@microsoft.com`

## Abstract

Complex systems are often decomposed into modular subsystems for engineering tractability. Although various equation based white-box modeling techniques make use of such structure, learning based methods have yet to incorporate these ideas broadly. We present a modular simulation framework for modeling *homogeneous* multibody dynamical systems, which combines ideas from graph neural networks and neural differential equations. We learn to model the individual dynamical subsystem as a neural ODE module. Full simulation of the composite system is orchestrated via spatio-temporal message passing between these modules. An arbitrary number of modules can be combined to simulate systems of a wide variety of coupling topologies. We evaluate our framework on a variety of systems and show that message passing allows coordination between multiple modules over time for accurate predictions and in certain cases, enables zero-shot generalization to new system configurations. Furthermore, we show that our models can be transferred to new system configurations with lower data requirement and training effort, compared to those trained from scratch.

## 1 Introduction

Modeling and simulation are key enablers in engineering as viable alternatives to real world systems. Simulation allows for the testing and evaluation of large/complex systems in a risk-free environment and has been of use in many disciplines. Simulation also plays a key role in domains such as deep reinforcement learning, which requires a large volume of interactions with the environment to learn effective policies. However, traditional simulators often require a large amount of engineering effort to create and can be computationally intensive when high fidelity predictions are needed. This has created interest in data-driven simulations, where machine learning algorithms are used to learn system dynamics directly from observations [1, 2, 3, 4, 5]. While machine learning algorithms excel at learning correlations from data, such end-to-end learning approaches result in models that cannot be reused for different system configurations. Additionally, end-to-end learning also requires large amounts of data and training effort when trying to model real-world systems.

In this paper we propose a machine learning based framework to learn and compose modular simulators. We build upon the observations that complex systems comprises of multiple modules (e.g., multibody systems), where each module evolves according to its own set of dynamical equations with corresponding inputs and outputs [6]. Examples include multiple pendulums connected via a single thread and multiple quadrotors carrying a large structure or a beam. Instead of modeling the entire system jointly, it might be more beneficial to model individual subsystems (such as a single pendulum or one quadrotor) and then consider interaction effects. The ability to harness such modularity can help reduce the sample complexity for machine learning as well create transferable subsystems that enable rapid prototyping.

---

\*Equal contributions.

36th Conference on Neural Information Processing Systems (NeurIPS 2022).

The technical methodology we describe here builds upon a class of models referred to as *mechanistic models*. Mechanistic models aim to introduce inductive biases related to the underlying mechanisms of the systems being modeled, which are in turn informed by domain knowledge. Some examples are neural networks motivated by differential equations [7, 8, 9, 10, 11, 12, 13], physics-informed neural networks [14, 15, 16, 17, 18, 19, 20] etc. As mechanistic models encode structural knowledge of the system at hand, they can be used to learn modular simulations, as opposed to treating the entire system as a singular block.

In this work, we propose an approach to simulate homogeneous systems from data. Under the assumption that the systems of interest can be modeled as ordinary differential equations, we present a class of neural ODEs which we call *message-passing neural ODEs* (MP-NODE). Instead of modeling the entire graph of a system through ODEs, the message passing neural ODE only learns the model of the subsystem that forms represents a node in the graph. In order to enable interaction between the nodes, the MP-NODE contains additional message variables along with the state variables. The entire system is thus composed of several MP-NODE modules with the weights shared between them, and the coordination as well as ability to account for model approximation errors over time is learnt through the messages. We show that MP-NODEs allow for expressive and efficient modular simulations which can be fine-tuned to new system configurations significantly faster than training a model from scratch. Our key contributions are listed below.

1. We present the message-passing neural ODE (MP-NODE), an augmented neural ODE formulation aimed at enabling interaction between multiple components through messages.

2. We test the MP-NODE on various connected systems and demonstrate its ability to learn complex dynamics and generalize to unseen system configurations.

3. We also show that this modular framework allows finetuning for new configurations with significantly lower training effort than training models from scratch.

## 2 Related Work

Let us assume $X_t \in \mathcal{X}$ is the state of the world at time $t$. Then, dynamically stepping over $T$ timesteps with optional control input $U_t \in \mathcal{U}$ gives a trajectory of states $\mathbf{X}_{t_{0:T}} = (X_{t_0}, \ldots, X_{t_T})$. A *simulator* $s : \mathcal{X} \times \mathcal{U} \to \mathcal{X}$, models the dynamics by mapping previous states and control inputs to future states. Traditional simulators often define the dynamics as a modular set of differential or differentiable algebraic equations and use appropriate numerical solvers as the update mechanism.

For data-driven simulations, we parameterize the dynamics of the simulator as $f_\theta : \mathcal{X} \times \mathcal{U} \to \mathcal{Y}$. Here, the parameters $\theta$ can be optimized for some training objective. The semantics of the dynamics information $Y \in \mathcal{Y}$ is governed by an update function, usually an integrator.

Learning such simulations from data has been quite important in fields like physics [21] and graphics [22]. Learned simulations can often be faster and more efficient for complex phenomenon prediction than just equation based simulators [23, 24].

Combining graph neural networks (GNN) [25, 26, 27] and differential equations [7, 8] has been explored in Poli et al. [28], Huang et al. [29] and Zang and Wang [30]. The dynamics of their test systems are relatively simple. Importantly, there is no study of transferability of the learned dynamics to changing graph structure. More recently, graph neural networks have been applied in the context of particle simulations by Sanchez-Gonzalez et al. [31] and mesh-based simulations by Pfaff et al. [32]. Moreover, unlike other graph neural net based dynamics learning works [28, 30] they also use edge based features. However, these largely ignore the existence of control inputs, limiting their usefulness for reinforcement learning and control applications.

Some work in the past has examined the idea of message passing for continuous time Bayesian networks [33], or as a combination of generative graphical models and GNNs [34]. The focus of these works was to model full system dynamics for a particular graph structure, where messages play a role in computing the interactions, but do not evaluate transfer performance to different graph structures. We distinguish our work from such prior methods primarily through the fact that our method is generalizable between different graph structures. An approach that is close to ours from the neural ODE body of work would be the augmented neural ODEs[8] which also contain extra dimensions in the state, although these additional variables are zero-valued. In our method, we repurpose those extra dimensions to explicitly pass messages across multiple neural ODEs as per the graph structure.

# 3 Methodology

The proposed framework tackles the problem of learning modular simulations where (1) smaller subsystems are individually modeled and (2) can be composed to simulate larger systems and processes. Specifically, given observations from a complex system, the methodology seeks to learn both the individual simulation module as well the message passing mechanism that would best explain the data traces.

At the core of this method are Neural Differential Equations that use messages to coordinate amongst themselves. Such message passing enables the framework to model various interaction effects between the modules as well as account for errors due to model approximations and time discretization. The next subsection reviews Neural ODEs. We then present our MP-NODE which extends neural ODEs to incorporate message passing. Note that we assume that the graphs are homogeneous and the underlying graph structure is given. However, we do show that the learnt Neural ODEs can be reused across different given graph topologies. Handling heterogeneous subsystems along with learning the underlying graph structure is a more challenging problem and will be the focus of future work.

## 3.1 Background: Neural Differential Equations

Many continuous-time dynamical systems which can be found in physics (oscillators, mechanical systems), biology (population dynamics) etc. are best modeled by ordinary differential equations. Typically, such a system can be described using an ODE as $\dot{x} = f(x, u, t)$. A data-driven way of learning a simulation of this system would thus be to estimate $\theta$ such that $\dot{x} = f_\theta(x, u, t)$ fits a set of given observations.

Neural ordinary differential equations [7] are a class of neural networks that link the concepts of residual networks and dynamical systems. Given a set of observations of the system state over time, under the assumption that the system dynamics can be described through an ODE, neural ODEs can be used to approximate the time derivative directly. By integrating the model output using a black-box ODE solver for an arbitrarily long timestep and backpropagating appropriately, the model parameters $\theta$ can be updated to minimize the distance between the predicted states (Equation 2) and the observed ones.

$$\frac{d\hat{\mathbf{x}}(t)}{dt} = \mathbf{f}_\theta\left(\mathbf{x}(t), \mathbf{u}(t), t\right) \tag{1}$$

$$\hat{\mathbf{x}}(t+1) = \mathbf{x}(t) + \int_t^{t+1} \mathbf{f}_\theta\left(\mathbf{x}(t), \mathbf{u}(t), t\right) dt \tag{2}$$

## 3.2 Message Passing with Neural ODEs (MP-NODE)

Now, let us consider a connected system that evolves as $\frac{dX}{dt} = f(G, X, U, t)$, where $X(t) \in \mathbb{R}^{n \times d}$ represents the entire system state, constituting of $n$ connected nodes whose state is of length $d$ each. $G = \{\mathcal{V}, \mathcal{E}\}$ captures the graph structure and how the nodes are connected to each other, which we assume is known. $\mathcal{N}(k)$ refers to the neighbors of a node $k$ in the graph.

While neural ODEs have been successfully used to model dynamical systems, modeling large systems with complex connectivity and dynamics is still challenging and computationally intensive. Furthermore, even the slightest change to the connectivity of the system necessitates retraining of the entire system dynamics model. On the other hand, connected systems can often be decomposed into repeatable subsystems that communicate with each other, hence it can be more efficient to learn these individual blocks rather than attempting to model the connected system as a whole.

The first key aspect of our framework is that the focus is placed on the subsystems rather than the entire system. For a given homogeneous network, we essentially only learn the model corresponding to a single subsystem, or node. When orchestrating full simulation of the system, the weights are shared across all individual nodes.

Secondly, due to this modular nature, the individual nodes need to learn to coordinate with each other and understand how their long term evolution changes based on interaction according to the graph structure. We facilitate such communication between the nodes by augmenting each subsystem state with additional variables which we call *messages* with the augmented state for node $k$ being

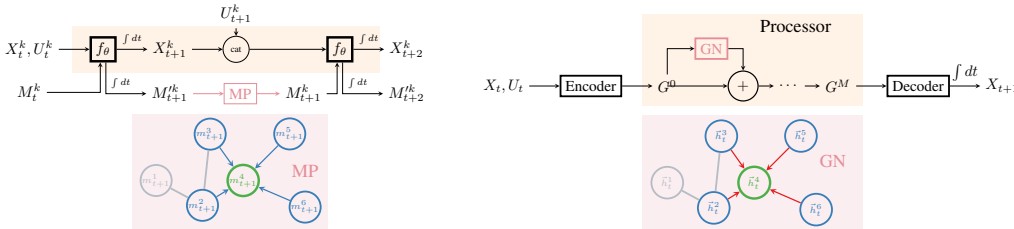

Figure 1: (Left) MP-NODE architecture: Individual nodes can simulate independently and coordinate via message passing. (Right) Learning to Simulate (L2S) architecture without edge features.

$\mathbf{X}^k = \begin{bmatrix} \mathbf{x}^k \\ \mathbf{m}^k \end{bmatrix}$. With the neural ODE operating on the augmented state, each neural subsystem outputs a message along with its predicted state at each timestep. Similarly, each node also receives a message at the input, which results from an aggregation operation between all output messages of the node neighbors which are identified according to $G$. We call this model the *message passing neural ODE* (MP-NODE) and describe the structure in Figure 1(Left). The full formulation of the MP-NODE can thus be written as below, where $\mathbf{m}'_k(t)$ is the input message at node $k$ and $\mathbf{m}_k$ is the output message.

$$\mathbf{m}'_k(t) = \text{MEAN}\left(\{\mathbf{m}_j(t), j \in \mathcal{N}(k)\}\right) \tag{3}$$

$$\frac{d}{dt}\begin{bmatrix} \mathbf{x}_k(t) \\ \mathbf{m}_k(t) \end{bmatrix} = \mathbf{f}_\theta\left(\begin{bmatrix} \mathbf{x}_k(t) \\ \mathbf{m}'_k(t) \end{bmatrix}, \mathbf{u}(t)\right) \tag{4}$$

$$\hat{\mathbf{X}}_k(t + dt) = \text{ODESOLVE}\left(\mathbf{f}_\theta\left(\begin{bmatrix} \mathbf{x}_k(t) \\ \mathbf{m}'_k(t) \end{bmatrix}, \mathbf{u}(t), dt\right)\right) \tag{5}$$

While the outputs from the ODE solver contain both the predicted state and a message, the supervision and loss computation only happen on the predicted states, allowing the messages to evolve freely. Over time, the model is able to use these additional variables to potentially encode the effects of different aggregations and interactions between neighbors, finally reaching an optimum configuration that is applicable for all the nodes. Learning such a generalizable subsystem allows the MP-NODE framework to be directly applicable for another system configuration with higher or lower number of nodes, as opposed to attempting to learn dynamics for the entire system state such as is done with a conventional neural ODE, which needs retraining for new configurations.

The length of the message can be varied, and we treat it as a hyperparameter that changes the expressivity of the model. In our implementation, we use the mean as the aggregator for incoming messages from the neighbors of a node.

## 4 Experiments

We evaluate the performance of MP-NODE on five different systems: two of them selected to focus on robotic regimes and three of them as representative of standard graph neural network benchmarks, for all of which we generate datasets of trajectories. We implement our method in Julia [35] and make use of the SciML ecosystem [36, 37]. We list training-specific details such as hyperparameters, learning rates, optimizers in the appendix. Source code for our experiments is available at `https://github.com/microsoft/MPNODE.jl`.

We list the systems used below, and provide additional details in the appendix.

- *Coupled Pendulum:* This system is a combination of two identical simple pendula connected by a string, behaving in an undamped manner. Hence, we train an MP-NODE that models a single pendulum, which is instantiated twice and the communication is enabled through the messages.

- *Lorenz Attractor:* A Lorenz system is a simplified model of atmospheric convection, described through a 3-dimensional system that exhibits chaotic behavior in specific cases [38]. In our case, we consider multiple Lorenz systems coupled in a fully connected manner. We denote Lorentz3 and Lorentz10 as fully connected systems with 3 and 10 nodes respectively.

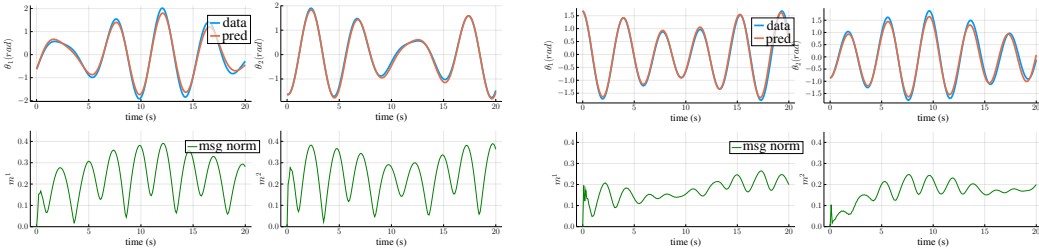

Figure 2: The figure compares the predicted (red) with observed (blue) state evolution of the coupled pendulum system. In the first row, the two pairs of columns are two different instance of trajectories. Each column represent one pendulum and the green curves represents outgoing messages. The messages enable interaction between the connected pendulums and enable accurate modeling.

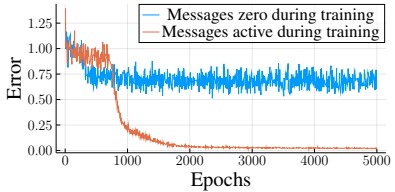
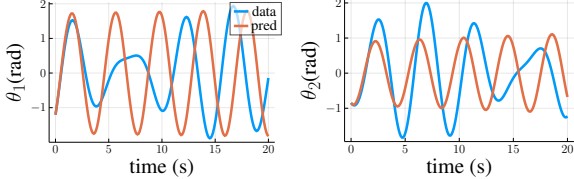

(a) Comparison of normal messages vs. set to zero during training. Training loss is much lower when messages are active (red) vs. when they are set to zero (blue).

(b) When message passing is disabled at test time, each node acts similar to an unconnected, simple pendulum. The two graphs represent the two nodes and we plot the prediction (red) and the observed (blue) state evolution.

Figure 3: MP-NODE applied to the coupled pendulum system. Together with Figure 2 we show how the messages help minimize drift and compare the characteristics of the system when messages are disabled.

- *Gene Dynamics:* This system simulates a continuous time system from biology: the gene regulatory dynamics governed by the Michaelis-Menten equation[39] over grid-like configurations with varying connectivity. Similar to one of the test cases in [30], we generate data for three types of graph topologies: a) Erdós-Rényi (ER) [40]; b) Barabási-Albert (BA) [41] and c) Wattz-Strogatz (WS) [42]. We use two configurations in our experiments, where the dynamics are simulated over a $4 \times 4$ grid and an $8 \times 8$ grid.

- *Kuramoto System:* The Kuramoto model [43] describes the behavior of large sets of coupled oscillators. We simulate the Kuramoto dynamics over the same three types of networks mentioned above, for 10 nodes, which we refer to as Kuramoto10. We use Lindner et al. [44] to build our reference simulators to generate our datasets.

- *Quadrotor swarm:* This system involves a more challenging dynamics learning task of multiple quadrotors carrying a heavy load (that a single quadrotor cannot lift) while passing through a narrow doorway [45]. We consider two versions of this: one with 3 quadrotors, another with 6.

We compare our proposed approach to the one proposed by Sanchez-Gonzalez et al. [31] that generalized different graph based dynamics learning methods, which we refer to as the L2S baseline. We do not use edge features in our implementation of L2S (Figure 1(Right)).

## 4.1 Effect of messages

When the MP-NODE method of modeling is applied to systems with connected nodes, we find that over time, the messages evolve to enable coordination between the nodes. Figure 2 compares predictions (red) with observations (blue) of state evolution for the coupled pendulum system. Each column corresponds to a pendulum and we plot two different trajectories (rows 1 and 3) in Figure 2. We observe that the predictions closely match the ground truth observations. We also plot the norm of outgoing messages (green) (rows 2 and 4) in Figure 2 and notice that they dynamically change in order to appropriately coordinate across the two nodes.

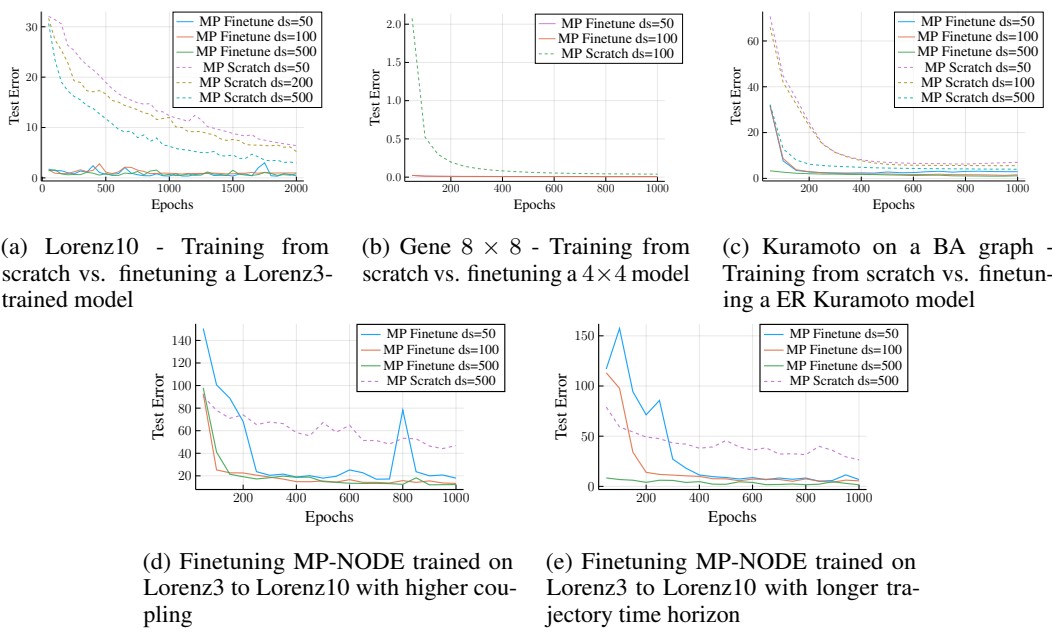

(a) Lorenz10 - Training from scratch vs. finetuning a Lorenz3-trained model

(b) Gene $8 \times 8$ - Training from scratch vs. finetuning a $4 \times 4$ model

(c) Kuramoto on a BA graph - Training from scratch vs. finetuning a ER Kuramoto model

(d) Finetuning MP-NODE trained on Lorenz3 to Lorenz10 with higher coupling

(e) Finetuning MP-NODE trained on Lorenz3 to Lorenz10 with longer trajectory time horizon

Figure 5: MP-NODE performance when trained from scratch vs. finetuning an existing model

We also investigate the importance of message passing by comparing performance of MP-NODE with and without messages. Specifically, we turn the messages off by clamping the message outputs to zero. Note that such forcing the messages to zero is similar to the augmented ODE applied to individual nodes. Figure 3a shows that we achieve lower training loss with messages on (red) vs when off (blue). We again test the idea of forcing messages to zero during training (similar to augmented neural ODE) on the Kuramoto system. As can be seen from Figure 4, using the additional dimensions for message-passing between nodes as in MP-NODE leads to substantial improvements in learning performance. This is consistent with the findings for the coupled pendulum system.

Similarly, Figure 3b highlights that if no messages are passed between the nodes at test time, each node starts

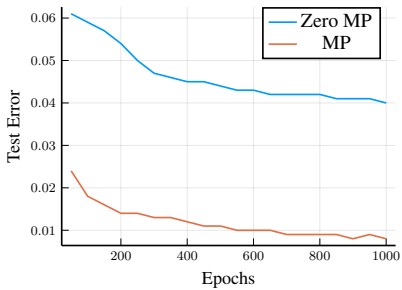

Figure 4: Performance on Kuramoto System: Normal MP-NODE vs. MP-NODE with messages set to zero

behaving like a simple pendulum that is unconnected to any neighbors. This indicates that the message parameters are important and help the model by encoding the interaction between the nodes.

## 4.2 Finetuning

The modular nature of the MP-NODE also allows for easy transfer to different configurations of systems. We examine the ability and performance of finetuning in several cases.

**Increasing number of nodes:** A common use case would be to take a trained MP-NODE module, representing a subsystem, and finetune it for a larger graph than what it was originally trained on. We use the MP-NODE that was trained on Lorenz3, and finetune this model for the Lorenz10 configuration. We compare this training effort with that of MP-NODE models trained from scratch just for Lorenz10. We show in Figure 5a that even with a small dataset of 50 trajectories, finetuning a pretrained MP-NODE reaches a low error significantly faster than the models trained from scratch with larger training sets.

Similarly, we put the modular nature of the MP-NODE to test in the gene dynamics system by taking MP-NODE models trained with smaller grid sizes and finetuning them on data from larger grid sizes. In Figure 5b, we show the finetuning performance when a model trained on a $4 \times 4$ grid is transferred

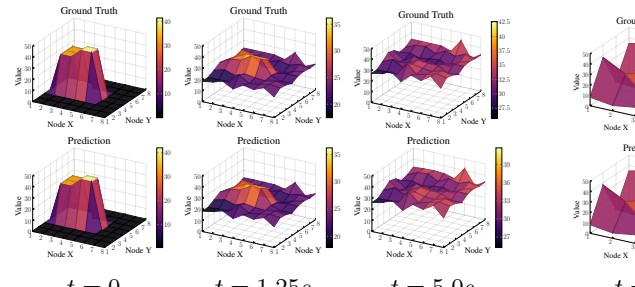
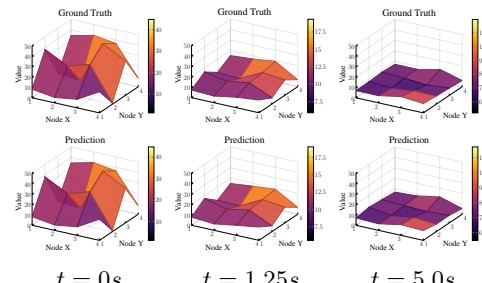

(a) MP-NODE predictions on a $8 \times 8$ grid of gene regulatory dynamics finetuned over a model trained on $4 \times 4$ grid data.

(b) MP-NODE predictions on a $4 \times 4$ grid of gene regulatory dynamics using the ER network, a topology unseen during training.

Figure 6: Performance of the MP-NODE on the gene dynamics model: finetuning to larger grids, as well as zero-shot generalization to unseen graph structures.

Table 1: Zero-shot generalization performance of MP-NODE

| Validation | | Test | | |
|---|---|---|---|---|
| BA | BA - L2S | BA | ER (unseen) | WS (unseen) |
| $0.0051 \pm 0.0001$ | $0.0106$ | $0.0045 \pm 0.0004$ | $0.0026 \pm 0.0008$ | $0.0023 \pm 0.0002$ |

to an $8 \times 8$ grid, compared to one trained from scratch on the $8 \times 8$ grid. Similar to the trend with the Lorenz systems, we see that finetuning a trained MP-NODE allows for accurate predictions faster than training a model from scratch. Furthermore, in Figure 6a, we show a qualitative comparison of ground truth and predicted state over the finetuned Gene $8 \times 8$ grid at different time steps.

This shows that due to the modular characteristic of our approach, finetuning a pretrained MP-NODE module allows for rapid adaptation to arbitrary number of nodes in a graph.

**Higher system complexity:** We also evaluate the possibility of finetuning MP-NODEs for different system parameters. As an example, we finetune a MP-NODE model trained on Lorenz3 to Lorenz10 but with different coupling intensity and show the results in Figure 5d. We observe that finetuning requires a lot less data to achieve better performance than training from scratch. Similarly, when we finetune the MP-NODE model trained on Lorenz3 to Lorenz10 for longer time horizon of $10s$ in Figure 5e, we again find that a lot less data is required to achieve better performance than training from scratch.

**Changing graph structure:** We also examine the ability of MP-NODE to be finetuned for different graph structures. To this end, we train an MP-NODE on a Kuramoto10 system connected according to the Barabasi-Albert (BA) network and attempt to finetune it for systems of other network types. Similar to above, we see that finetuning is more efficient at adapting to new network types than training models from scratch. A comparison of training effort is shown in Figure 5c when transferring the Kuramoto10 model trained on the BA network to the ER network.

### 4.3 Zero-shot generalization

Considering that the modular nature of MP-NODE allows us to connect arbitrary number of subsystems together for a given graph structure, we can investigate the performance for new configurations without explicitly finetuning the model. When tested on the Lorenz system, we find that an MP-NODE trained only on Lorenz3 exhibits reasonable zero-shot generalization performance on a higher number of Lorenz attractors, such as Lorenz7 and Lorenz10 (see appendix) without requiring any additional training.

We notice that this ability holds for not only changing numbers of nodes, but also different graph topologies. We train the MP-NODE on the Gene $4 \times 4$ system connected according to the Barabasi-Albert (BA) topology, with the dataset comprising of data corresponding to five different adjacency matrices in the BA topology. By training on data coming from different types of connections, the

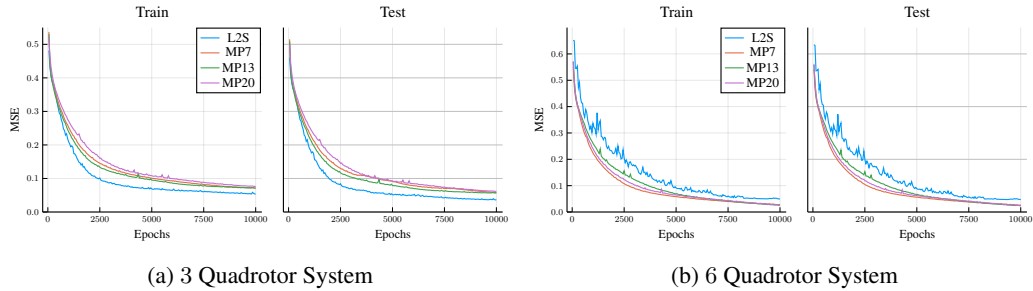

(a) 3 Quadrotor System                (b) 6 Quadrotor System

Figure 7: Effect of message dimensionality on train/test performance of the MP-NODE examined for Quadrotor systems

MP-NODE learns how node dynamics are are affected by different graph structures. After training, we observe that the MP-NODE model that was trained only on BA topology generalizes to the other unseen topologies as well (ER and WS). Table 1 shows the MP-NODE's performance on both seen and unseen network topologies. We observe that MP-NODE exhibits low test error on unseen topologies as well, while achieving better validation performance than L2S. In Figure 6b, we take a qualitative look at the predictions for gene dynamics on the ER network using the zero-shot transferred model.

## 4.4 Effect of message size

In order to further improve the performance when modeling generally complex dynamics of systems such as Kuramoto or the quadrotor swarm, we investigate whether having a larger message size helps. We train MP-NODE models on the Kuramoto10 system with message dimensionalities of $1, 3, 7,$ and $13$. By default, each Kuramoto node has a state length of 3. We show this comparison in Figure 8. We notice that larger messages result in higher accuracy, but after a certain threshold (messsage size $= 7$ in this case), the improvement becomes minimal.

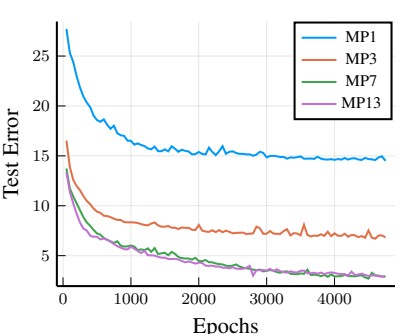

Figure 8: Effect of message size on test performance in Kuramoto system

In the case of the quadrotor swarm, we choose message sizes of $7, 13, 20$ for analysis. Through Figure 7, we observe that all tested message sizes for MP-NODE lead to similar final test errors. Although L2S seems to do slightly better on the 3 Quadrotor System, it performs worse on the 6 Quadrotor System.

Similar to the trend in other datasets, we find in Figure 9 that finetuning MP-NODE on even limited data allows learning more accurate simulations much faster than learning from scratch, and this holds true for different message sizes as well. Although, we note that higher message dimensions can lead to higher data requirements.

## 4.5 Summary

To better demonstrate the advantages of MP-NODE when it comes to finetuning for different system configurations, we consider a metric: for a given test configuration, we compute the 'number of epochs taken to reach minimum test error multiplied by the value of the minimum test error observed'. Intuitively, we can see that a lower number is better. We summarize our findings in Table 2, where we compare MP-NODE finetuning vs. training from scratch, and for some configurations, we also compare with L2S. In the case of Lorenz and Kuramoto systems, we evaluate the performance under varying number of trajectories in the finetuning dataset. Through Figure 9, we observe that MP-NODE finetuning consistently achieves a lower combination of test error and number of epochs required to reach it, even in the low-data regime.

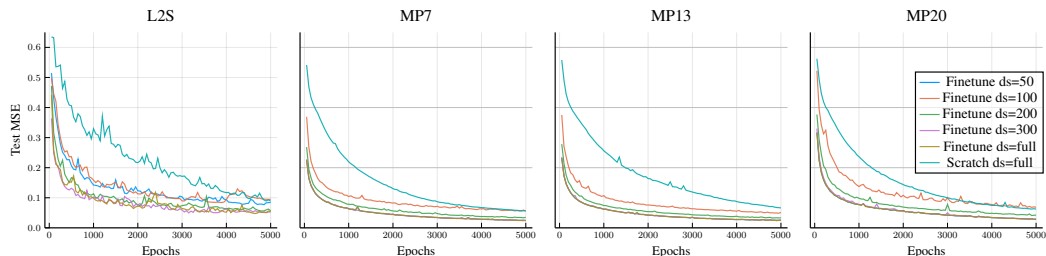

Figure 9: Performance of a 3 Quadrotor MP-NODE finetuned on the 6 Quadrotor System

Table 2: Summary of the performance of MP-NODE finetuning vs. other baselines, using the metric of minimum test error multiplied by number of epochs taken to reach the minimum. Lower is better.

| Model | Lorenz10 | | | Kuramoto | | | Gene Dynamics | Quadrotors |
|---|---|---|---|---|---|---|---|---|
| | ds=50 | ds=100 | ds=500 | ds=50 | ds=100 | ds=500 | | |
| MP-NODE finetuned | **632.70** | **833.91** | **477.75** | **1358.50** | **1264.83** | **874.80** | **12.53** | **236.00** |
| MP-NODE scratch | 12800 | 10454 | 5656 | 5008 | 5268 | 3974 | 28 | N/A |
| L2S | N/A | N/A | N/A | N/A | N/A | N/A | 46.64 | 476.0 |

# 5   Conclusions

We present a framework for modular modeling of complex homogeneous systems called the message passing neural ODE (MP-NODE). MP-NODE builds upon neural ODEs, and learns a model for the individual node within a connected homogeneous system, which can then be shared across all instances in the system. Augmenting the system state with additional message variables allows the model to encode the features required for coordination and accurate prediction over long horizons. This modular way of thinking about complex systems opens up several possibilities such as zero-shot generalization to unseen configurations and rapid finetuning to more complex configurations. We apply our framework to a wide variety of systems such as coupled pendulum, Lorenz attractors, biological networks, Kuramoto systems and a swarm of quadrotors, and showcase its ability to learn complex dynamics in a modular fashion while outperforming existing methods. We also show that pretrained MP-NODE models can be finetuned for new configurations significantly faster than training from scratch which has the potential to greatly amortize data and compute requirements for data-driven simulations. While we focused on neural ODEs, general principles of parameter sharing and message passing as used in this work would be applicable to modern state-space models [46] too.

**Limitations:**   Current implementation of MP-NODE is limited to homogeneous systems only. For better ability to adapt to more complex systems, an extension that can handle heterogeneous systems would be needed. Although encoder-decoder architectures could help in extending to nodes with different node state dimensions, different time-stepping requirements for the nodes might require a different messaging setup. Moreover, the approach currently requires full knowledge of the graph structure which may not always be available for large complex systems. In our current problem setup, we also do not consider potential issues such as delays in communication between nodes, which we leave for future work.

**Broader Impact:**   Pretrained simulation nodes that are reusable can minimize data, compute, and energy requirements. Data-driven simulations can often speed up conventional slow simulations while also being capable of capturing real world phenomena ignored in traditional simulations. We show through our analysis of finetuning and generalization that our approach of modeling subsystems that are inherently reusable as opposed to specific configurations of systems has the potential to alleviate data and compute requirements. However, limited data and lack of physical constraints can lead to arbitrarily bad predictions for unseen simulation regimes. Nevertheless, building upon this approach, there is a potential to enable large variety of existing simulations to work with heterogeneous compute hardware and data via surrogate modeling [47].

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
