# A   Experiment Details

Source code for the training pipeline, tasks, and models used in this work, is available as part of the supplementary material.

We used the same $Adam$ [48] optimizer for all our experiments and a learning rate of $0.001$, and a batch size of 128. For solving the differential equations both during ground truth data generation as well as with the neural ODEs, we use the Tsitouras 5/4 Runge-Kutta (Tsit5) method from DifferentialEquations.jl [36].

## A.1   Coupled Pendulum

The coupled pendulum dynamics are defined as

$$\ddot{\theta}_1 = \frac{\sin\theta_1 * (m_1 l_1 \dot{\theta}_1{}^2 - g - kl_1) + kl_2 \sin\theta_2}{m_1 l_1 \cos\theta_1} \qquad (6)$$

$$\ddot{\theta}_2 = \frac{\sin\theta_2 * (m_2 l_2 \dot{\theta}_2{}^2 - g - kl_2) + kl_1 \sin\theta_1}{m_2 l_2 \cos\theta_2}$$

Where $\theta_i$, $\dot{\theta}_i$ refer to the angle and the angular velocity of the $i^{\text{th}}$ pendulum respectively. $m_i$ and $l_i$ are the mass and length corresponding to the $i^{\text{th}}$ pendulum, which are chosen to be 1.0kg and 1.5m respectively. $k$ is the spring constant of the coupling string (chosen to be 2.0), and $g$ is the gravitational acceleration. The system state is defined as $[\theta_1, \dot{\theta}_1, \theta_2, \dot{\theta}_2]$.

We train the MP-NODE on a dataset of 500 trajectories, each randomly initialized with state values between $[-\pi/2, \pi/2]$ for the $\theta$ and $[-1, 1]$ for $\dot{\theta}$, with a time step of $0.1$s and each trajectory 10s long. The dataset is normalized through Z-score normalization. We use the mean squared error loss during training.

## A.2   Lorenz Attractor Systems

A Lorenz system is a simplified model of atmospheric convection, described through a 3-dimensional system that exhibits chaotic behavior in specific cases. The system has a three-dimensional state whose evolution is described as $\dot{x} = \sigma(y-x); \dot{y} = x(\rho-z)-y; \dot{z} = xy-\beta z$. In our implementation, we use the values of $\sigma = 10.0, \rho = 28.0, \beta = 2.666$ for the Lorenz attractor which match the values originally used in [38]. We consider multiple Lorenz systems coupled in a fully connected manner.

We generate data from multiple Lorenz attractor configurations, the variables being the number of Lorenz nodes $(3, 7, 10)$, the length of the trajectories (2.5s, 5s) and the coupling magnitude. We generate the coupling matrix with random coupling strengths between 0 and 1, with the diagonal elements set to zero (as the nodes are not connected to themselves). We use an additional coupling magnitude parameter that the coupling matrix is multiplied with to strengthen/weaken the coupling effect: which is set to $0.01$ for low coupling and $1.0$ for high coupling.

We train our MP-NODE primarily on the Lorenz3 system, with trajectories that are 2.5s long with a dt (timestep) of 0.05s and low coupling magnitude. We perform Z-score normalization on the data and use Huber loss along the time dimension as the training loss function. This model is later finetuned on the other configurations.

## A.3   Gene Dynamics

The gene regulatory dynamics for a grid-based system are governed by the Michaelis-Menten equation that is shown below [39].

$$\frac{dx_i(t)}{dt} = -b_i x_i{}^g + \sum_{j=1}^{n} A_{ij} \frac{x_j{}^h}{x_j{}^h + 1} \qquad (7)$$

We solve the Michaelis-Menten equation over two sizes of 2D grids: $4 \times 4$ (gene_small) and $8 \times 8$ (gene_large). The adjacency matrices were generated randomly according to three network topologies:

a) Erdós-Rényi (ER) [40]; b) Barabási-Albert (BA) [41] and c) Wattz-Strogatz (WS) [42]. For each grid with $n$ cells in total, each cell is connected to $n/2$ other cells. The initial state at each cell of the grid was chosen randomly to be between $0$ and $50$. Our implementation loosely follows that of [30].

The training dataset contains 200 trajectories from gene_small with five different adjacency matrices of the power-law network type, of which 70% are used for training and the rest for evaluation. Each trajectory is 5s long with a timestep of 0.1s. For training the MP-NODE on this system, we use the mean squared error loss. This model is later finetuned on gene_large, and the other network topologies.

## A.4 Kuramoto Systems

The Kuramoto model [43] describes the behavior of large sets of coupled oscillators. Variations of the Kuramoto model find applications in a variety of fields, such as neuroscience [49], power systems [50] and vehicle coordination [51]. The dynamics are defined as:

$$\frac{dx_i(t)}{dt} = b_i + \sum_{j=1}^{N} A_{ij} \sin(x_j - x_i) \tag{8}$$

We also simulate the Kuramoto systems with adjacency matrices according to three network topologies: random, power-law and small-world. We generate data for Kuramoto systems with 10 nodes, with each node connected to 5 other nodes. The initial state at each cell of the grid was chosen randomly to be between -1 and 1. The timestep for the trajectories was set to $0.05$s.

We train the main MP-NODE model on a dataset of 500 trajectories, of which 70% are used for training and the rest for evaluation. We perform Z-score normalization on the data and use Huber loss along the time dimension as the training loss function. This model is later finetuned on data from other network topologies.

## A.5 Load Carrying Quadrotors

Quadrotor dynamics are defined by:

$$\dot{x} = \begin{bmatrix} \dot{r} \\ \dot{q} \\ \dot{v} \\ \dot{\omega} \end{bmatrix} = \begin{bmatrix} v \\ \frac{1}{2} q \otimes \hat{\omega} \\ g + \frac{1}{m} \left( R(q) F(u) + F_c(u_5, x, x^\ell) \right) \\ J^{-1} \left( \tau(u) - \omega \times J\omega \right) \end{bmatrix} \tag{9}$$

where $r \in \mathbb{R}^3$ is the position, $q$ is the unit quaternion, $R(q) \in \mathbb{SO}(3)$ is quaternion dependent rotation matrix from body frame to world frame, $v \in \mathbb{R}^3$ is the linear velocity in the world frame, $\omega \in \mathbb{R}^3$ is the angular velocity in the body frame. $g$ is the gravity vector, $m$ is the mass of the individual quadrotor, $J \in \mathbb{S}^3$ is the moment of inertia tensor, $q_2 \otimes q_1$ denotes quaternion multiplication, and $\hat{\omega}$ denotes a quaternion with zero scalar part and $\omega$ vector part. The forces ($F \in \mathbb{R}^3$) and torques ($\tau \in \mathbb{R}^3$) are in the body frame. For our experiments, the coupling between the quadrotors was provided via (unobserved) state vector of the load $x^\ell \in \mathbb{R}^6$. Therefore the state vector $x \in \mathbb{R}^{13}$ and the control vector $u \in \mathbb{R}^5$. For more details please refer to [45].

The training dataset was generated using the batch trajectory optimizer in [45]. We generated a total of $463$ trajectories for the 3 quadrotor system with different initial conditions of the load and the quadrotors. We used 70% trajectories for full training and rest for evaluation. The trajectories were 10s long with timestep, $dt = 0.2$s. Similar method was used to generate data for 6 quadrotor system.

The data was standardized using the Z-score transform. Mean-Square-Error between the predicted trajectory and the ground truth trajectory was used as the training objective.

# B Additional Experiments

## B.1 Coupled Pendulum

We show coupled pendulum dynamics evolution and the corresponding messages from a different initial state in Figure 10.

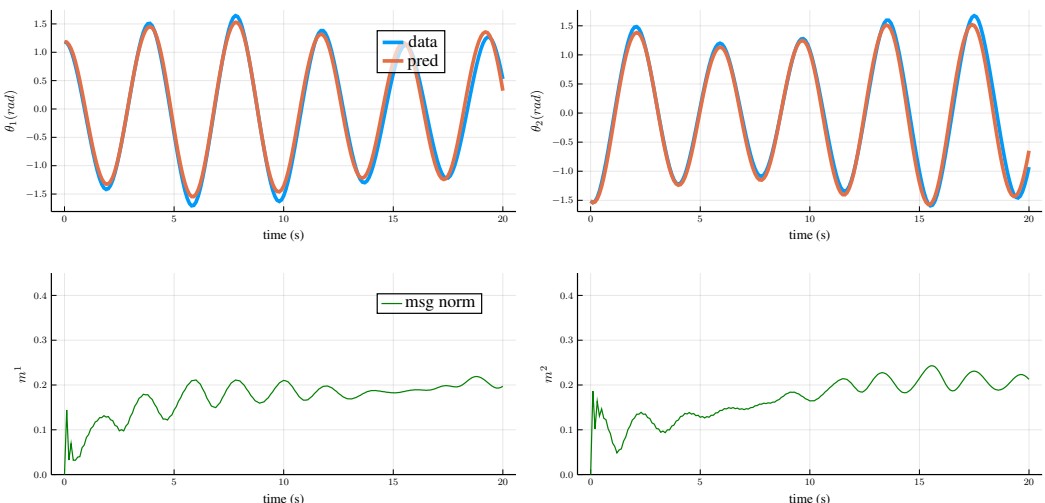

Figure 10: Additional example of the state of each pendulum and corresponding messages that allow for accurate predictions.

## B.2 Lorenz Systems

In section 4.3, we discuss the zero-shot generalization ability of the MP-NODE. In Figure 11a and Figure 11b, we show results of a MP-NODE model trained only on Lorenz3, but tested on Lorenz7 and Lorenz10 without any finetuning. We observe reasonably good performance on Lorenz7 and 10 which are unseen configurations during training.

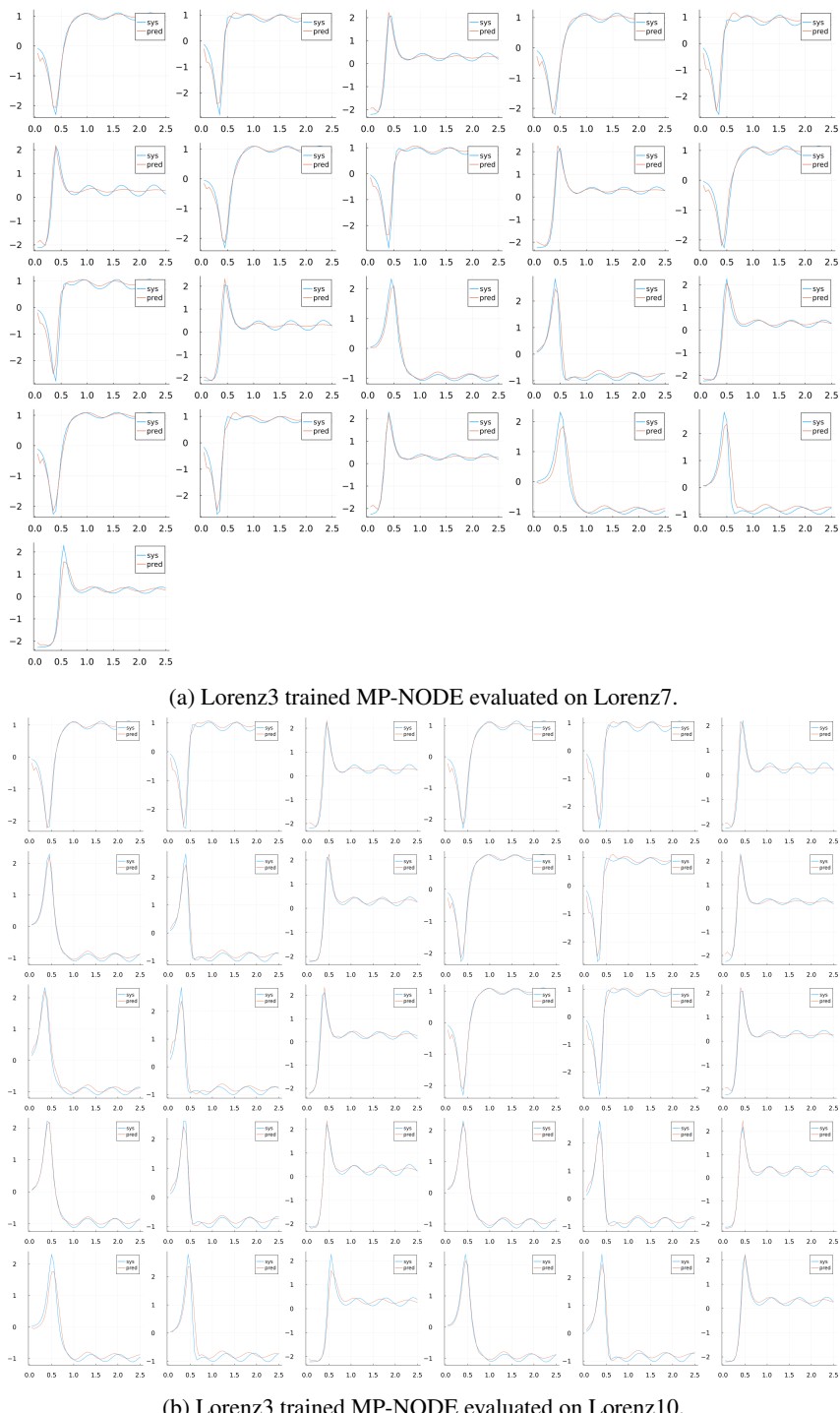

(a) Lorenz3 trained MP-NODE evaluated on Lorenz7.

(b) Lorenz3 trained MP-NODE evaluated on Lorenz10.

Figure 11: Zero-shot generalization of MP-NODE on Lorenz systems. From left to right, top to bottom - plots depict the predicted vs. ground truth dynamics on X, Y, Z states of all 7 or 10 nodes.

## B.3   Quadrotor Systems

With increasing dimensions of messages, it is difficult to analyze the roles they play. To this end we apply Principal Component Analysis (PCA) for all 39 messages over time for 100 trajectories and visualize the evolution of first three principal components over the time horizon of the trajectory in Figure 12. There is an overall trend of increasing values of these components over time. We believe that these increasing values hint at the messages acting in to counteract the problem of simulation drift over time. The pattern of smaller values at some time indices likely corresponds to the nature of dynamic interaction when the quadrotors reconfigure themselves to carry the load through the narrow doorway.

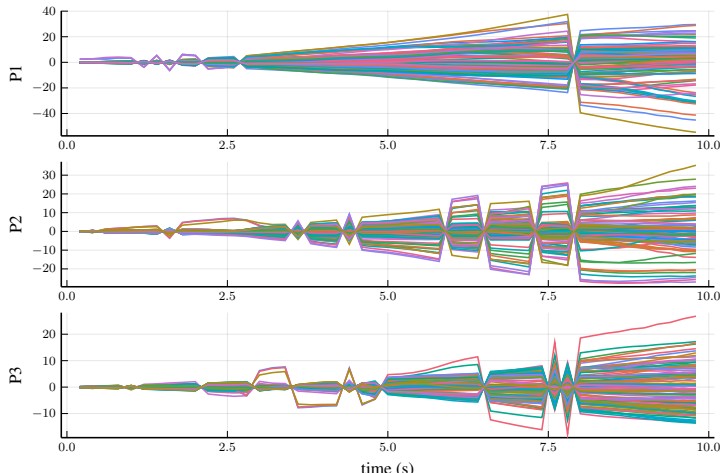

Figure 12: First three principal components of messages over time for 100 different trajectories.