# OpenReview forum: "Learning Modular Simulations for Homogeneous Systems"
_NeurIPS.cc/2022/Conference — NeurIPS 2022 Accept_

### Official Review · Reviewer_jBWD · 2022-07-11

**Rating:** 8
**Confidence:** 3
**Soundness:** 3 good
**Presentation:** 4 excellent
**Contribution:** 3 good

**Summary:**

In this paper, the authors tackle the problem of generating data from a modular system, where each submodule is assumed to be identical, e.g. they follow the same dynamics. The authors introduce a novel extension of neural ODES, which they call message-passing neural ODES (MP-NODE),  where the state is augmented with messages, allowing for the individual submodules to interact with one another.

**Questions:**

I have a condition concerning the initial conditions for the messages. How are they set? And how does the initial condition affect the performance of the method?

**Limitations:**

The limitation section was very honest which I very much appreciated!

**Strengths And Weaknesses:**

# Strengths

MP-NODE is a simple and elegant solution, which I am a huge fan of. The inductive bias of the proposed approach is fantastic as assuming that each submodule follows the same ODE allows for weight sharing, substantially reducing the number of parameters **and** easily allows for generalization. In fact, this is the coolest thing to me; that you can easily deploy the method on a system with more nodes than it was trained on *without* increasing the number of parameters!

I also thought the large variety of experiments was very compelling and really showed the possibilities of this approach over a wide range of domains.

# Weaknesses

The one disappointing thing is the lack of error bars! Especially given how close some of these training curves are, errors bars are essential!

---

> ### Author Response · Authors · 2022-08-02
> **Response to Reviewer jBWD**
>
> We thank the reviewer for their inspiring review.
> ### Lack of error bars
> We found our current supervised learning regime to have much smaller variance than other learning regimes like RL. Most of the learning curves that are close, are ablations of the proposed method itself. Gap is maximum during initial training epochs and lower data availability as one would expect. Nevertheless, we will try to rerun the experiments with multiple seeds for the camera-ready version.
> ### Initial messages
> We set our initial messages to zero as we assume that there is no prior communication between the nodes at the beginning. In principle we could try random initialization, but zero-init sufficed for our usecase.

---

> > ### Comment · Reviewer_jBWD · 2022-08-08
> > **response**
> >
> > Thanks for responding! While I'm sure the variance was small, for reproducibility and fairness of comparisons, error bars are essential! Moreover, they inspire confidence in the approach as it makes it less likely that the random seed was optimized for instance.

---

### Official Review · Reviewer_X5Qn · 2022-07-12

**Rating:** 6
**Confidence:** 3
**Soundness:** 3 good
**Presentation:** 3 good
**Contribution:** 3 good

**Summary:**

This paper proposes a message-passing neural ODE method to learn the simulations of homogeneous systems. In this method, smaller sub-systems are individually modeled by a neural ODE. The same weights are shared among each sub-system. A message-passing mechanism is introduced to model the interactions between sub-systems.

Experiments were made, including:
- Test and evaluate the method on five different systems. The method shows good results in these systems.
- Experiments about finetuning (and zero-shot generalization) the model to a system with more nodes, higher complexity, and changed graph structures. And the method shows good generalization abilities.
- Experiments about the effects of message passing and message size.
- Comparison with the L2S baseline (proposed by Sanchez-Gonzalez et al. [31])


**Questions:**

Did you specifically choose "Tsitouras 5/4 Runge-Kutta (Tsit5)" as the integration method for solving ODEs. Do you think using other methods would significantly affect the results or not?

Different timesteps (0.05s, 0.1s, 0.2s) were used in different experiments. Were there specific reasons to choose these values? Would the method stay robust if different (especially larger) timesteps are used in the training data? Would larger timesteps cause significant delays in message passing?

In the finetune results (figure 3), for some cases (e.g., figure 3a, d, e), finetune has much better results than training from scratch. Is the model much harder to converge when the system contains more sub-systems? If a smaller system is not given, but the model needs to directly learn on a relatively large system, can it still learn well?

In "Table 2", is "Gene Dynamics" 4*4 or 8*8, is "Quadrotors" 3 or 6-quadrotor system?

In line 270, the number of epochs taken to reach minimum test error is used in the evaluation metrics. For the proposed method and L2S baseline, do they take a similar or very different amount of times for one single epoch?




**Limitations:**

The authors discussed the limitations of this paper:
- The method can only be applied to homogeneous systems;
- The underlying graph structure of the system needs to be given;
- Potential issues such as delays in communication between nodes are not considered in this paper.


**Strengths And Weaknesses:**

Strengths
The proposed method shows novel points compared to the regular neural ode methods and the regular graph neural network methods.
The authors made a fair amount of experiments to show the accuracy of this method, the effects of message passing, and the generalizability of the proposed model.

Weaknesses
The systems that this paper makes experiments on are a little simple and small - although the underlying ode can be complex, each sub-system usually contains only several degrees of freedom (e,g, only several particles), and the number of sub-systems in a whole system is also relatively small. On the contrary, as mentioned in the paper's "related work" section, there are other learning-based simulation methods that have made successful results on more complex systems, such as clothes, fluids, etc. The authors may at least have some discussions about whether (and how) this paper can be applied to more complex systems.

---

> ### Author Response · Authors · 2022-08-02
> **Response to Reviewer X5Qn**
>
> We thank the reviewer for their detailed review.
>
> ### Scaling to more complex systems
> Systems can be complex along different dimensions: variety of nodes, number of nodes, complexity of individual node dynamics. Complexity with respect to systems such as clothes, fluids is relatively easy to handle because they are often modeled as large numbers of homogenous particles and it is just a matter of scaling our method on larger GPUs; individual components like message passing on graphs, integrator time-stepping etc. already work on GPUs. However experiments with respect to generalization and transfer as we do here, are more difficult to setup for such systems.
> As we mention in our limitations, we are still limited to homogenous systems and increased complexity from heterogeneity would require more future work.
>
> ### Choice of integration scheme
> We chose Tsit5 because it’s the default recommendation for non-stiff equations in DifferentialEquations.jl library. The choice might matter when dealing with stiff systems. However, our current systems of interest were not stiff. Otherwise the choice of integration scheme should not matter much for our time horizons.
> ### Choice of different timesteps
> We mostly chose the timesteps as used by previous works that introduced the domain. Given the variety of timesteps across different domains, we believe that our method is largely robust to such choices. However beyond a certain threshold of sampling frequency (related to Nyquist criterion) we start losing dynamics information in the dataset itself which would affect learning performance.
> ### Finetuning performance
> Training performance on a larger system from scratch depends on the domain dynamics complexity but most importantly data availability. Larger systems would require a lot more data to converge to equivalent performance when trained from scratch.
>
> ### Table clarifications
> The numbers listed in table 2 refer to the finetuning performance: for the Gene Dynamics system, finetuning was evaluated on the 8x8 grid, and for quadrotors, it was evaluated on the 6 quadrotor system.
> ### Relative time for a single epoch
> Since our method passes messages in time, computing gradients and therefore training is slower than baseline L2S method which only does message passing between nodes in the current timestep. However, during rollout there is little time difference

---

### Official Review · Reviewer_Qynb · 2022-07-12

**Rating:** 6
**Confidence:** 4
**Soundness:** 2 fair
**Presentation:** 3 good
**Contribution:** 3 good

**Summary:**

The paper proposes to model the evolution of coupled homogeneous dynamical systems by combining neural ODE and message passing in graph neural networks. A NODE is used to model a single subsystem and message passing is used to describe the interactions of the various subsystems. Simulation results are reported on a number of bechmarks.

**Questions:**

- I checked the appendix but I did not find the setting for the parameters of the Lorenz system (and the others). Maybe I missed this detail, but this is a very relevant detail. In fact, the Lorenz system can behave in very different ways (e.g. show chaotic behavior or not) depending on the setting of its parameters. This will have a significant effect on the time horizon of the predictions and it is something that must be discussed.

- Are 2.5 seconds enough to fully explore the Lorenz attractor? Can you report some more details/plots showing the quality of the predictions?

**Limitations:**

The main and obvious limitation is the focused applicability to coupled homogeneous systems. This aspect has been clearly discussed in the paper.

**Strengths And Weaknesses:**

Strengths:
- The paper is well-written and easy to follow
- The idea is intriguing and with wide applicability

Weaknesses:
- Experimental results are not compared against baselines. There is a some comparison on 2 systems with the L2S system, but I am not sure why they did not report results on all systems. This makes it difficult to assess the quality of the results.
- Some more technical details on the dynamical systems setting are needed.
- The methodological novelty is clearly limited (from a technical point of view), as the contribution consists of combining two existing methodologies.

---

> ### Author Response · Authors · 2022-08-02
> **Response to Reviewer Qynb**
>
> We thank the reviewer for their comments and questions.
> ### Baseline comparisons
> We omitted L2S comparison in certain experiments because it failed to perform well to do meaningful comparisons.
> ### Methodological novelty
> While we agree that our method is quite simple and from a certain perspective a combination of existing things, no other method has demonstrated similar transfer capabilities yet. We believe our method will be a useful baseline for future explorations in this domain, while being easy to train given its similarity to the Neural ODE.
>
> While we used a generic parametrization for an individual node, in principle domain dependent physics informed parametrization could also be used within the same pipeline.
>
> ### Parameter settings for Lorenz System
> While these details were there in the code in the supplementary material, we have also added these details to the appendix. For easier reference, the values used are $\sigma=10, \rho=28, \beta=8/3$ as in the original formulation of the Lorenz attractor [1]. Sample rollouts from this configuration of the Lorenz attractors can be seen in Figure 8 in the Appendix. With this default configuration, we noticed that 2.5 seconds was sufficient to capture the main behavior of the attractor, after which the nodes settle into a steady state oscillation. Other choices could result in significantly more chaotic behavior that would require further analysis and evaluation.
>
> [1] Lorenz, Edward N. "Deterministic nonperiodic flow." Journal of atmospheric sciences 20, no. 2 (1963): 130-141.

---

### Meta-Review · Area_Chair_wjbL · 2022-08-26

**Recommendation:** Accept
**Confidence:** Certain

**Metareview:**

The manuscript describes a novel combination of message passing / graph neural network ideas with NODE. Reviewers agree that this is an innovative contribution. Authors have provided various non-trivial experiments to demonstrate its performance.

**Award:**

No

---

### Decision · Program_Chairs · 2022-09-14

Accept